# Acute Kidney Injury Associated with Severe SARS-CoV-2 Infection: Risk Factors for Morbidity and Mortality and a Potential Benefit of Combined Therapy with Tocilizumab and Corticosteroids

**DOI:** 10.3390/biomedicines11030845

**Published:** 2023-03-10

**Authors:** Jose Iglesias, Andrew Vassallo, Justin Ilagan, Song Peng Ang, Ndausung Udongwo, Anton Mararenko, Abbas Alshami, Dylon Patel, Yasmine Elbaga, Jerrold S. Levine

**Affiliations:** 1Department of Medicine, Jersey Shore University Medical Center, Neptune, NJ 07753, USA; 2Department of Nephrology, Community Medical Center, RWJBarnabas Health, Toms River, NJ 08757, USA; 3Department of Medicine, Hackensack Meridian School of Medicine, Nutley, NJ 07110, USA; 4Department of Pharmacy, Community Medical Center, RWJBarnabas Health, Toms River, NJ 08757, USA; 5Department of Medicine, Community Medical Center, RWJBarnabas Health, Toms River, NJ 08757, USA; 6Hackensack Meridian School of Medicine, Nutley, NJ 07110, USA; 7Department of Pharmacy, Monmouth Medical Center Southern Campus, RWJBarnabas Health, 600 River Ave., Lakewood, NJ 08701, USA; 8Department of Medicine, Division of Nephrology, University of Illinois Chicago, Chicago, IL 60612, USA; 9Department of Medicine, Division of Nephrology, Jesse Brown Veterans Affairs Medical Center, Chicago, IL 60612, USA

**Keywords:** COVID-19, acute kidney injury, anti-inflammatory, Tocilizumab

## Abstract

Background: Acute kidney injury (AKI) is a common complication in patients with severe COVID-19. Methods: We retrospectively reviewed 249 patients admitted to an intensive care unit (ICU) during the first wave of the pandemic to determine risk factors for AKI. Demographics, comorbidities, and clinical and outcome variables were obtained from electronic medical records. Results: Univariate analysis revealed older age, higher admission serum creatinine, elevated Sequential Organ Failure Assessment (SOFA) score, elevated admission D-Dimer, elevated CRP on day 2, mechanical ventilation, vasopressor requirement, and azithromycin usage as significant risk factors for AKI. Multivariate analysis demonstrated that higher admission creatinine (*p* = 0.0001, OR = 2.41, 95% CI = 1.56–3.70), vasopressor requirement (*p* = 0.0001, OR = 3.20, 95% CI = 1.69–5.98), elevated admission D-Dimer (*p* = 0.008, OR = 1.0001, 95% CI = 1.000–1.001), and elevated C-reactive protein (CRP) on day 2 (*p* = 0.033, OR = 1.0001, 95% CI = 1.004–1.009) were independent risk factors. Conversely, the combined use of Tocilizumab and corticosteroids was independently associated with reduced AKI risk (*p* = 0.0009, OR = 0.437, 95% CI = 0.23–0.81). Conclusion: This study confirms the high rate of AKI and associated mortality among COVID-19 patients admitted to ICUs and suggests a role for inflammation and/or coagulopathy in AKI development. One should consider the possibility that early administration of anti-inflammatory agents, as is now routinely conducted in the management of COVID-19-associated acute respiratory distress syndrome, may improve clinical outcomes in patients with AKI.

## 1. Introduction

Severe COVID-19 disease, as caused by SARS-CoV-2 infection, results in a wide variety of renal injuries, manifesting as hematuria, proteinuria, tubular dysfunction, acute tubular necrosis, thrombotic microangiopathy (TMA), and/or collapsing glomerulopathy [1,2,3]. The incidence of acute kidney injury (AKI) with COVID-19 disease ranges from 0.5% to 36% among all infected patients and is much higher in patients requiring intensive care management [4,5,6,7,8,9,10]. The reasons for the variability in the development of AKI are not completely clear, and may be related to timing of diagnosis, severity of illness, differences in demographics, and definitions of AKI [10,11]. Aside from the development of COVID-19-associated respiratory distress syndrome (C19-ARDS), AKI in the setting of COVID-19 disease is the second most common cause of morbidity and mortality [5,12,13]. The prevalence of AKI in non-survivors of COVID-19 having C19-ARDS and infected with the Wuhan strain of the virus during the first wave of the pandemic was increased 2.5-fold compared to survivors of COVID-19 with C19-ARDS [12].

Although COVID-19 disease primarily impacts the respiratory tract, the enzyme angiotensin converting enzyme-2 (ACE-2), the primary receptor for SARS-CoV-2, is expressed by the endothelium of almost all extra-pulmonary organs, including the reproductive tract. Interaction of SARS-CoV-2 with the ACE-2 receptor leads to a pro-thrombotic hyper-inflammatory state as well as maladaptive hyper-activation of the innate immune system resulting in cytokine storm. In this respect, COVID-19 disease can be viewed as a multisystem disease, with both acute and chronic multiorgan dysfunction, including myocarditis, myocardial infarction, macrophage activation syndrome, arterial and venous thrombosis, encephalitis, stroke, thyroiditis, adrenal insufficiency, hepatitis, hepatic failure, preeclampsia, and sterility [14,15,16,17,18].

Moreover, infection with COVID-19 is associated with several pathophysiologic phases: an incubation phase, an early viremic phase (which may or may not be symptomatic), and an early inflammatory pulmonary phase occurring between seven and fourteen days after infection [19]. In some patients, this is followed by a progressive inflammatory pulmonary phase leading to the development of C19-ARDS and multiorgan dysfunction [19]. An imbalance among pro- and anti-anticoagulant factors, in part a result of hyper-inflammation and cytokine storm, produces a pro-thrombotic state characterized by disseminated intravascular coagulation, macro- and microvascular thrombotic events, and multiorgan failure [19].

A myriad of insults, each capable of causing AKI, can occur during severe COVID-19 infection. These include hyper-inflammation as a part of cytokine storm, septic-shock-associated hypoperfusion, ischemia, C19-ARDS-associated hypoxia, and vascular thrombosis [1,2,20,21,22]. Previous studies of COVID-19 patients found that the risk factors for developing AKI were similar to those identified in other groups of patients with critical illness, such as comorbidities, age, vasopressor requirement, and respiratory failure requiring mechanical ventilation [2,3,21,22,23]. What remains unclear is the impact and interaction of inflammatory-coagulopathic biomarkers and other clinical and laboratory variables on the development and mortality of COVID-19-associated AKI. In addition, few studies have included therapeutic agents among the variables examined. With this in mind, we evaluated the incidence, risk factors, and medical therapy associated with the development and outcome of AKI in patients with severe COVID-19 disease admitted to an intensive care unit (ICU).

## 2. Materials and Methods

To determine risk factors for the development of AKI in patients with severe COVID-19 infection, and to compare risk factors for mortality between patients with and without AKI, we conducted a retrospective analysis of 249 consecutive patients either admitted or transferred to the ICU of two community hospitals between 12 March 2020 and 17 June 2020. At the time of ICU entry, all patients manifested severe difficulty in breathing as evidenced by one or more of the following: respiratory rate greater than 30 breaths per minute, blood oxygen saturation of 93% or less on room air, PAO2/FIO2 ratio less than 300, and presence of lung infiltrates in more than half of the lung fields [24,25].

Inclusion criteria were age greater than 18, confirmed diagnosis of COVID-19 by a positive PCR test, and signs and symptoms of COVID-19 infection. Study baseline was the time of hospital admission. Patients with end-stage renal disease (ESRD) were excluded from the analysis. All patients received standard-of-care therapy. Management and timing of ventilator support, employment of C19-ARDS ventilator strategies, antibiotic use, antiviral therapy, anticoagulation, initiation of vasopressors, use of convalescent plasma, corticosteroid (CC) therapy, and Tocilizumab therapy were determined by the ICU physician and consultants.

All standard laboratory results were extracted from the electronic medical record and were performed according to standardized laboratory practices. D-Dimer levels were obtained using a latex agglutination photo-optical assay from a sample of citrated whole blood. C-reactive protein (CRP) levels were obtained from serum samples and analyzed incorporating a laser-nephelometric method. Missing values in the case of standard laboratory results were not imputed and accounted for less than 1% of all measurements. Missing biomarker data, including CRP, D-Dimer, and ferritin, were found to be missing completely at random (MCAR) by Little’s MCAR test. Substituted values were inserted using an expectation maximization algorithm [26,27]. Missing biomarker data are reported in Appendix A.

Demographics, comorbidities, and clinical outcome variables were obtained from the electronic medical record or the patient’s history and physical exam and entered into a de-identified database. Measurements obtained on admission included arterial blood gas (ABG), routine metabolic chemistries, CRP, D-Dimer, ferritin, WBC and differential, and all variables necessary to calculate a Sequential Organ Failure Assessment (SOFA) score on admission. Other collected data included the dates of admission, ICU transfer, and death; need for vasopressor therapy; need for mechanical ventilation; PAO2/FIO2 ratio; use of corticosteroid (CC) therapy; and use of azithromycin, hydroxychloroquine (HCQ), convalescent plasma, Tocilizumab, or heparin (low molecular weight or unfractionated) either as deep vein thrombosis (DVT) prophylaxis or anticoagulant therapy. If patients received both anticoagulant doses and thrombo-prophylactic doses of heparin, they were analyzed in the anticoagulant group.

AKI was defined according to Kidney Disease: Improving Global Outcomes (KDIGO) criteria, namely, an increase in serum creatinine value (SCr) by ≥0.3 mg/dL (≥26.5 μMol/L) within 48 h. Urine output was not considered. SCr on admission was used to assess the change in SCr [28,29,30]. For this study, AKI associated with COVID-19 is defined as AKI occurring during the first 7 days of admission. The consulting nephrologist determined the timing and indication for the initiation of renal replacement therapy (RRT).

All statistical analyses were performed with IBM SPSS 28 software (IBM SPSS Inc., Chicago, IL, USA). We performed both univariate and multivariate analyses. The primary outcome was development of AKI for patients admitted or transferred to the ICU during the index admission. Summary statistics were computed for patients who did or did not develop AKI and for survivors vs. non-survivors, as well as for various subgroups of those who did and did not develop AKI. Continuous variables were expressed as median with interquartile ranges (IQR) and compared by the Student’s *t*-test or the Wilcoxon rank-sum test, as appropriate. Categorical variables were compared with Pearson’s chi-squared test. Fisher’s exact test was employed when indicated. Variables that were significant by univariate analysis at *p* < 0.05 were candidates for multivariate analysis. To determine risk factors that were independently associated with the outcome of AKI, we performed multivariate analysis by logistic regression with stepwise forward variable selection. Goodness of fit was determined by the Hosmer–Lameshow test. For continuous variables, the odds ratio (OR) represents the relative amount by which the odds ratio for the outcome variable increases or decreases when the independent variable is changed by exactly one unit. ORs and their 95% confidence intervals (95% CI) were determined by exponentiation of the beta coefficient and its upper and lower CI, respectively. If variables included imputed values for missing measurements, we performed multivariate analysis with and without these variables. Cox proportional hazards with forward variable selection was performed to determine variables independently predictive of the outcome of survival among all patients, and in those who did or did not develop AKI. For continuous variables, the hazard ratio (HR) represents the relative amount by which the probability of obtaining the outcome variable increases or decreases when the independent variable is changed by exactly one unit. HRs and their 95% CIs were determined by exponentiation of the regression coefficient and its upper and lower CI, respectively.

## 3. Results

### 3.1. Patient Characteristics

To evaluate risk factors for the development of AKI and its associated mortality, we retrospectively evaluated severe COVID-19 admissions (those patients requiring ICU admission or transfer). From 12 March 2020 to 17 June 2020, a total of 249 patients met the definition of severe COVID-19 infection; 90 (36%) were admitted directly to the ICU, and 159 (64%) were admitted to COVID-19 units and subsequently transferred to the ICU. The median age was 70 years (IQR 61, 80). There were 148 males (59%), and 177 patients were Caucasian (71%). The majority of patients had severe respiratory failure requiring mechanical ventilation (n = 173, 69%).

The mortality rate among these patients was 65% (162 patients). There was no significant difference in mortality between those directly admitted to the ICU (n = 56, 62%) vs. those transferred to the ICU (n = 106, 66%) (*p* = 0.48, OR = 0.82, 95% CI = 0.48–1.40). Likewise, there was no significant difference in the development of AKI between those directly admitted to the ICU (n = 40, 44%) and those transferred later (n = 79, 50%) (*p* = 0.43, OR = 0.8, 95% CI = 0.48–1.36). The median time to ICU transfer was three days (IQR 1, 5). For those patients not initially admitted to an ICU, there was no statistically significant difference in median time to ICU transfer between survivors and non-survivors (2 days (IQR 1, 4) vs. 3 days (IQR 1, 6), *p* = 0.1). Similarly, there was no significant difference in median time to ICU transfer between those who did and did not develop AKI (3 days (IQR 1, 6) vs. 3 days (IQR 1, 5), *p* = 0.17). As expected, patients directly admitted to the ICU had higher SOFA scores (6, IQR 3, 19) compared to those initially admitted to COVID-19 units (3, IQR 2, 5) (*p* = 0.00001).

### 3.2. Univariate Analysis of Risk Factors for AKI

Overall, 119 (48%) patients developed AKI, and 23 (19%) of these required RRT by the 7th day following hospital admission. Four additional patients developed AKI and required RRT later during their hospitalization (10–15 days). As the later development of AKI suggested an etiologic role for hospital-related factors distinct from initial COVID-19 infection, these patients were included in the non-AKI group. However, the inclusion of these late AKI patients in either group had no effect on the study’s results. The admission characteristics of the entire group of patients with severe COVID-19 infection are given in Table 1. Univariate analysis revealed that older age, the need for mechanical ventilation, elevated SCr, and higher SOFA score were associated with the development of AKI (Table 1).

Therapeutic and pharmacologic interventions are given in Table 2 and Figure 1. In general, information on the timing, rationale, and route of administration of agents was available only for CC and Tocilizumab and is therefore not a part of the analysis. Among all severe COVID-19 patients, 152 (61%) required vasopressor support, 192 (77%) received HCQ, 87 (35%) received azithromycin, 26 (10%) received Tocilizumab alone without CC, 69 (27%) received CC alone without Tocilizumab, 92 (37%) received combination therapy with both Tocilizumab and CC, and 65 (26%) received convalescent plasma. Consistent with admission during the first wave of the pandemic, only six patients (2.4%) received Remdesivir. Full-dose heparin was administered in 126 (50%) patients, and thrombo-prophylactic doses were administered in 85 (34%) patients. By univariate analysis, the need for vasopressors (*p* = 0.009, OR = 2.58, 95% CI = 1.25–5.31) and the use of azithromycin (*p* = 0.04, OR = 1.74, 95% CI = 1.03–2.95) were associated with the development of AKI. Conversely, patients who received combination therapy with Tocilizumab and CC were less likely to develop AKI (*p* = 0.018, OR = 0.53, 95% CI = 0.31–0.90). The use of either Tocilizumab alone or CC alone as monotherapy was not associated with a reduced risk of AKI. Only the combination predicted a lower risk of AKI. There was no difference in the time of initiation of CC among patients with and without AKI (4 days (IQR 1, 14) versus 3 days (IQR 1, 8), *p* = 0.56). Similarly, there was no difference in the time of initiation of Tocilizumab among patients with and without AKI (4 days (IQR 2, 6) versus 3 days (IQR 1.5, 7), *p* = 0.96).

Finally, since endothelial dysfunction, thrombosis, and inflammation are common features of COVID-19 disease, we examined the levels of inflammatory and coagulopathic markers in these patients [31,32]. These data were available from the electronic record, as it was common practice during the first wave of the pandemic to obtain serial inflammatory and thrombotic biomarkers in patients with COVID-19. Patients who developed AKI were noted to have significantly higher levels of D-Dimer on day 1 (hospital admission) and of CRP on day 2 (Table 3). These results are consistent with the interaction of SARS-CoV-2 with ACE2 on vascular endothelial cells in nearly all organs of the body, leading to excessive circulating levels of angiotensin 2, a pro-inflammatory, pro-thrombotic, and vasoconstrictive molecule [31,32,33].

### 3.3. Multivariate Analysis of Risk Factors for the Development of AKI

Independent predictors of AKI development were identified by stepwise logistic regression with forward variable selection for all predictive variables identified by univariate analysis (Table 4 and Figure 2). Increased risk of AKI was associated with elevated admission SCr (*p* = 0.0001, OR = 2.406, 95% CI= 1.56–3.70), vasopressor requirement (*p* = 0.0001, OR = 3.188, 95% CI = 1.69–5.98), elevated D-Dimer on day 1 (*p* = 0.008, OR = 1.0001, 95% CI = 1.000–1.0010), and elevated CRP on day 2 (*p* = 0.033, OR = 1.004, 95% CI = 1.0001–1.009). In contrast, co-administration of Tocilizumab and CC was independently associated with a lower risk for development of AKI (*p* = 0.009, OR = 0.437, 95% CI = 0.23–0.81). Elevated SCr, vasopressor requirement, and lack of co-administration of Tocilizumab and CC remained significant independent risk factors for AKI development even when the biomarker variables with missing and imputed values were removed from the analysis.

### 3.4. Univariate Analysis of Risk Factors for In-Hospital Mortality

We next examined the risk factors for in-hospital mortality among patients with severe COVID-19 infection. AKI development was associated with a 73% mortality, in contrast to a 56% mortality in patients not developing AKI. By univariate analysis, increased mortality was associated with older age, Caucasian race, presence of diabetes mellitus, presence of hypertension, development of AKI, requirement for RRT, need for mechanical ventilation, lower absolute lymphocyte count, elevated neutrophil to lymphocyte ratio, lower absolute platelet count, and elevated total bilirubin (Table 5).

To determine the effect of AKI on mortality, we compared the risk factors for mortality between patients who did and did not develop AKI. For patients with AKI, an increase in mortality was associated with need for mechanical ventilation, lower absolute lymphocyte counts, elevated absolute neutrophil count, elevated neutrophil to lymphocyte ratio, lower admission SCr, and lower absolute platelet count (Table 6). For those not developing AKI, increased mortality was associated with increased age, need for mechanical ventilation, presence of several co-morbidities (diabetes mellitus, hypertension, coronary artery disease, and chronic kidney disease (CKD)), increased admission SCr, and increased total bilirubin (Table 7).

The only pharmacologic intervention associated with increased mortality was the need for vasopressor support. This was true for the entire cohort (Table 8), as well as the subgroups with AKI (Table 9) and without AKI (Table 10). No inflammatory or thrombotic markers were associated with increased mortality (Table 11).

### 3.5. Multivariate Analysis of Risk Factors Associated with Mortality

Finally, to determine independent risk factors associated with mortality, we performed Cox proportional hazards analysis with forward variable selection (Table 12 and Figure 3). For the entire cohort of COVID-19 patients admitted to the ICU, only advanced age was independently associated with decreased survival (*p* = 0.00001, HR = 1.028, 95% CI = 1.016–1.041). For patients not developing AKI, increased age (*p* = 0.00001, HR = 1.044, 95% CI = 1.024–1.065) and CKD (*p* = 0.004, HR = 2.65, 95% CI = 1.37–5.1) were independently associated with decreased survival. For patients who developed AKI, surprisingly, the only independent risk factor was a lower SCr on admission (*p* = 0.011, HR = 0.79, 95% CI = 0.66–0.95).

## 4. Discussion

Clinical experience with patients hospitalized from COVID-19 has demonstrated significant heterogeneity in the development of AKI [10,34]. Chinese and Italian studies reported the development of AKI in 0.5–29% of patients [1,2,6,11]. In contrast, data from the United States and other developed nations have demonstrated a much higher incidence of this complication [9,12,35,36]. In a large cohort of COVID-19 patients, Fisher et al. found that COVID-19 patients developed AKI at twice the rate of historical controls [12]. The same investigators observed a similarly increased rate of AKI over the same timeframe when COVID-19 patients were compared to those admitted without COVID-19 [12]. Moreover, AKI in the setting of COVID-19 tends to be more severe. As compared to historical controls, COVID-19 patients had a 2.6-fold higher mortality and an increased requirement for RRT [37,38]. A recent meta-analysis involving over 25,000 COVID-19 patients found a pooled incidence of AKI of 53% in those with severe disease [39]. Schaubroeck et al., in a multi-center study involving 1286 critically ill patients, observed the development of AKI in 85% of patients [7]. In an ICU cohort of 313 ICU patients, Lumlertgul et al. observed the development of AKI in 76% of patients [40]. The results of our retrospective study are in accord with these data. AKI occurred in 119 of 249 (48%) of COVID-19 patients requiring ICU admission. Of these 119 patients, 23 (19%) required RRT, and 88 died (74%). These findings are consistent with previous studies from the US and other developed nations demonstrating an incidence of AKI in 40–80% of patients admitted to the ICU [5,7,12,40,41].

Despite these differences in incidence and severity, no risk factors have been consistently identified that discriminate between AKI developing in the presence or absence of COVID-19 infection. Traditional risk factors for AKI have included older age, elevated SCr on admission, requirement for vasopressors, thrombo-inflammatory biomarkers, and comorbidities such as diabetes and hypertension. By univariate analysis, in agreement with past studies, we found that the development of AKI was associated older age, the need for mechanical ventilation, elevated admission SCr, higher SOFA score, vasopressor requirement, and elevated inflammatory and thrombotic biomarkers (Table 1, Table 2 and Table 3). Of these, an elevated SCr, vasopressor requirement, and elevated inflammatory and thrombotic biomarkers were independent predictors (Table 4 and Figure 2). The lack of influence of comorbidities on the development of AKI may be a consequence of the advanced age of patients in our study (median age of 70), for whom comorbidities were similarly common in both AKI and non-AKI patients. In their studies of ICU patients with C19-ARDS, Wang et al. and Piniero et al. also found that the presence of multiple co-morbidities was not associated with the development of AKI [42,43].

A critical question is whether COVID-19-associated AKI is a separate clinical entity from other causes of AKI [44,45,46]. Investigations into this question have yielded conflicting results. Autopsy and biopsy series of COVID-19 patients with AKI have demonstrated a wide variety of glomerular and tubular injuries [44,46,47,48,49]. The prominence of glomerular and thrombotic injury, including collapsing glomerulopathy and TMA, is striking, and typically absent in sepsis-mediated AKI [46,47,49,50]. Nonetheless, notably, for both septic and COVID-19-associated AKI, morphologic and molecular evidence implicates inflammation as the primary driver of injury [51]. Thus, postmortem pathologic and molecular gene expression analyses have revealed that tubular injury occurring during the course of COVID-19 infection bears many similarities to sepsis-mediated acute tubular necrosis (ATN) [47,51]. Similarities include mitochondrial injury and dysregulation, increased autophagy, increased ceramide signaling, abnormalities in oxidative phosphorylation, up-regulation of apoptotic and necroptotic pathways, and increased endothelial and microvascular inflammation [51]. In addition, proteomic and genomic evidence indicates that a large number of genes involved in immune and metabolic counter-regulatory pathways, such as Treg differentiation and Sirtuin signaling, are expressed in both septic ATN and COVID-19-associated AKI [51].

In contrast, Volbeda and colleagues evaluated non-autolytic post mortem biopsies of patients with severe COVID-19 infection and observed morphologically more severe tubular injury, significant thrombosis of peritubular capillaries, and a less intense inflammatory response [52]. In fact, transcript levels for the pro-inflammatory cytokines TNF-α, IL-6, and MMP8 in biopsies from COVID-19-associated AKI patients were comparable to those found in normal controls [47,52]. In addition, the transcript levels of several genes, normally down-regulated during states of decreased renal perfusion, were also down-regulated in biopsies of COVID-19-associated AKI. Taken together, these differences suggest that decreased renal perfusion, rather than intense inflammatory state, may be the driver in some cases of COVID-19-associated AKI [52].

Such heterogeneity of pathologic and molecular findings may be due to differences in the timing of biopsies or possibly the existence of different endotypes within COVID-19-associated AKI. Certainly, the existence of an inflammatory phenotype is supported by the Recovery trial, which showed that the administration of Tocilizumab, a monoclonal antibody directed against the IL-6 receptor, led to a decrease in severe AKI requiring RRT [53]. Additional studies have demonstrated that the use of CC decreases the incidence of severe AKI [54,55,56]. Our own data also favor a role for inflammation in the pathophysiology of COVID-19-associated AKI. In the multivariate analysis, not only was increased CRP on day 2 independently predictive of AKI, but also combined treatment with Tocilizumab and CC was associated with decreased development of AKI. Our data further suggest a role for thrombosis, as an elevated level of D-Dimer on day 1 was also an independent predictor of AKI. In accord with this finding, Jewell et al. observed in a large UK cohort that patients who developed AKI had a statistically significant increase in thromboembolic complications [13]. Moreover, a systematic review has shown the risk of both venous and arterial thromboembolism to be statistically significantly higher in those with COVID-19 infection relative to those without [57].

With regard to mortality, although a large number of patient characteristics and interventions were associated with increased mortality, including the development of AKI, the only independent predictor for the entire cohort was advanced age. The impact of age is especially noteworthy, given that the median age of both survivors and non-survivors was greater than 65 and the difference in their median ages was less than 10 years (66 and 73, respectively). It is pertinent to note that the independent effect of older age on mortality is an almost universal finding. While one may associate increasing age with frailty and increased comorbidities, this may not be the entire explanation. For example, cellular expression of ACE-2, the putative receptor for the COVID-19 virus, decreases with age. As ACE-2 is capable of cleaving the pro-inflammatory peptide angiotensin 2 into the anti-inflammatory and anti-oxidant peptide angiotensin (1–7), aging may be associated with an increased tendency for inflammation [58,59,60,61]. Similar changes in ACE-2 expression have been observed in patients with cardiovascular disease [59,62,63,64,65]. Thus, aging and underlying cardiovascular disease may represent pro-inflammatory conditions predisposing patients to increased complications during the clinical course of COVID-19 disease.

Among patients without AKI, CKD was an additional independent predictor of mortality. Surprisingly, among patients with AKI, the only independent predictor of mortality was a lower SCr on admission. We can only speculate on the reasons for this anomalous finding. For example, a lower SCr on admission may have masked recognition of AKI and led to later renal consultation, resuscitative efforts, and use of RRT. Alternatively, a lower SCr might have been a surrogate for wasting or poorer nutritional status.

The retrospective and observational nature of our study necessarily entails several weaknesses. First, its retrospective nature limited our ability to determine and adjust for all differences in baseline characteristics and comorbidities. Second, we were unable to obtain information on such important potential risk factors as urinary output, urinary sediment, or proteinuria. Third, there were insufficient data to adjust for titration or timing and dose of pharmacologic interventions and therapies. Fourth, the diagnosis of comorbidities such as CKD or coronary artery disease were based on ICD coding rather than independent assessment by a non-involved clinician. Fifth, our retrospective analysis poses the usual limitations of selection and introduction bias. Thus, the benefits of anti-inflammatory therapy with Tocilizumab and CC in preventing AKI are associational and need to be confirmed in prospective trials. Sixth, there were many missing data points for the biomarkers D-Dimer, CRP, and ferritin, thus making it difficult to draw robust conclusions regarding their role as risk factors in the development of AKI. As a result, we had to rely on imputed data. Employing the imputed values into the multivariate analysis should therefore be considered exploratory. Nonetheless, removing the imputed biomarker data from the multivariate analysis did not alter the remaining risk factors. Further studies are needed to elucidate the role of biomarkers as predictors of AKI in patients with COVID-19 disease. Finally, our study was conducted during the first wave of the pandemic, before vaccinations were available, and extrapolation of our data to current patients may not necessarily be possible.

Despite these limitations, a major strength of our study is its study population, which comes from a large community hospital and is thus clinically and demographically similar to elderly COVID-19 patients in the general population.

## 5. Conclusions

The current study confirms the high rates of AKI and mortality among COVID-19 patients admitted to an ICU. As seen with other causes of AKI, multivariate analysis demonstrated that an elevated SCr on admission and the need for vasopressors were independent predictors of the development of AKI in patients with severe COVID-19. In addition, elevations of CRP and D-Dimer, biomarkers for inflammation and thrombosis, respectively, were also associated with an increased risk for AKI, albeit the effect was of small magnitude. Notably, patients who received both CC and Tocilizumab, but not either alone, were less likely to develop AKI. As has been shown for the development of C-19 ARDS in severely ill COVID-19 patients, our data suggest a potential role for inflammation in the development of COVID-19-associated AKI. One should consider the possibility that early administration of anti-inflammatory agents, just as one performs in the management of C-19 ARDS, may improve clinical outcomes in patients with AKI.

## Figures and Tables

**Figure 1 biomedicines-11-00845-f001:**
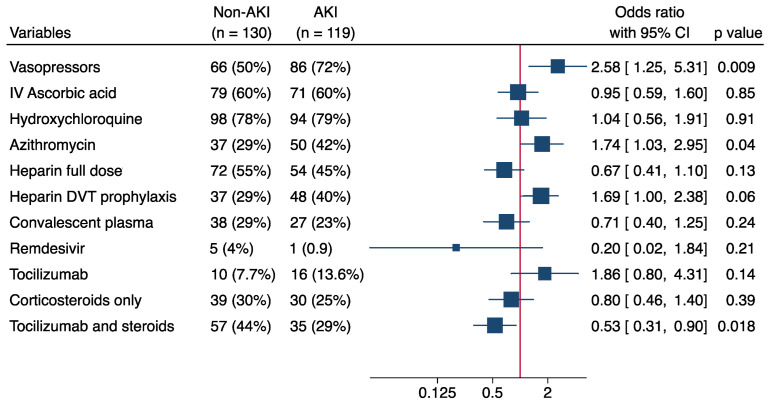
Forest plot depicting the effect of the indicated pharmacologic interventions on the odds ratio (OR) for developing AKI as determined by univariate analysis.

**Figure 2 biomedicines-11-00845-f002:**
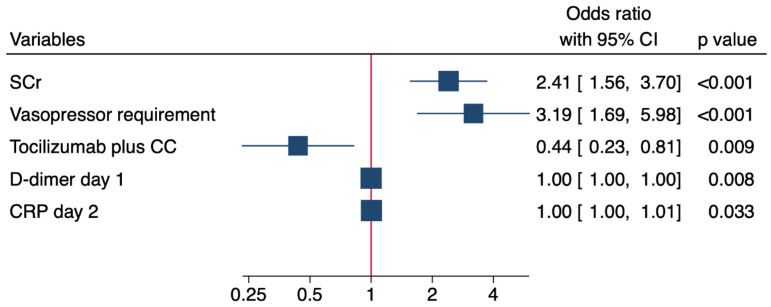
Forest plot depicting the impact of the indicated risk factors on the odds ratio (OR) for developing AKI as determined by logistic regression with stepwise forward variable selection. For continuous variables, the odds ratio (OR) represents the relative amount by which the odds ratio for the outcome variable increases or decreases when the independent variable is increased by exactly one unit.

**Figure 3 biomedicines-11-00845-f003:**
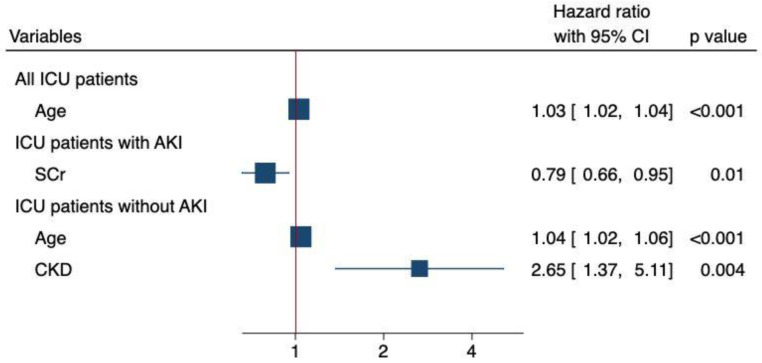
Forest plot depicting the impact of the indicated risk factors on the hazard ratio (HR) for mortality as determined by Cox proportional hazards with stepwise forward variable selection. For continuous variables, the hazard ratio (HR) represents the relative amount by which the probability of obtaining the outcome increases or decreases when the independent variable is changed by exactly one unit.

**Table 1 biomedicines-11-00845-t001:** COVID-19 ICU patients: characteristics of individuals with and without AKI.

	Non-AKI (n = 130)	AKI (n = 119)	*p*	OR	95% CI
Age	66 (58, 77)	73 (65, 82)	0.001		
Race (Caucasian)	88 (68%)	89 (75%)	0.22	1.41	0.81–2.46
BMI	29 (24.33)	29 (24, 35)	0.62		
Sex (male)	71 (55%)	77 (65%)	0.10	1.52	0.91–2.53
Diabetes	35 (27%)	41 (35%)	0.19	1.42	0.83–2.40
CHF	13 (10%)	17 (14%)	0.28	1.51	0.70–3.30
CAD	32 (25%)	29 (24%)	0.96	0.99	0.55–1.80
COPD	33 (25%)	27 (23%)	0.62	0.86	0.48–1.55
CKD	11 (8%)	15 (13%)	0.28	1.56	0.69–3.60
HTN	65 (50%)	70 (60%)	0.26	1.42	0.86–2.30
Cirrhosis	2 (1.5%)	2 (1.7%)	1.0	1.09	0.15–7.9
Malignancy	14 (11%)	13 (11%)	0.96	1.01	0.45–2.26
CVA	13 (10%)	14 (12%)	0.65	1.2	0.53–2.7
Mechanical ventilation	79 (61%)	94 (79%)	0.002	2.42	1.38–4.20
Neutrophiles × 10^9^/L	7.6 (4.4, 12)	7.6 (5, 13)	0.62		
Lymphocytes × 10^9^/L	0.8 (0.5, 1.3)	0.8 (0.5, 1.2.)	0.98		
Neutrophile/lymphocyte	8.8 (5.1, 16)	9.8 (5.2, 16)	0.64		
SCr (µmole/L)	84 (60, 127)	134.4 (97, 177)	0.00001		
Plts × 10^9^/L	246 (136, 308)	230 (164, 301)	0.44		
Tbili (µmole/L)	9.41 (6.8, 13.7)	8.5 (6.8, 15.4)	0.97		
SOFA admit	3 (2, 6)	5 (3, 8)	0.00001		
PaO2/FIO2	200 (100, 286)	201 (94, 314)	0.95		
Pa02	9.4 (7.4, 12.4)	10 (8.6, 11)	0.64		
FI02	0.36 (0.21, 1)	0.38 (0.23, 1)	0.72		

Abbreviations/legend: AKI = acute kidney injury, BMI= body mass index, CAD = coronary artery disease, CHF = congestive heart failure, CI = confidence interval, CKD = chronic kidney disease, COPD = chronic obstructive pulmonary disease, HD = hemodialysis, HTN = hypertension, OR = odds ratio, PaO2/FiO2 = ratio of partial pressure of oxygen to inspired concentration of oxygen, Plts = platelets, SCr= serum creatinine, SOFA = Sepsis-Related Organ Failure Assessment, TBili = total bilirubin.

**Table 2 biomedicines-11-00845-t002:** Pharmacological interventions in COVID-19 ICU patients with and without AKI.

	Non-AKI (n = 130)	AKI (n = 119)	*p*	OR	95% CI
Vasopressors	66 (50%)	86 (72%)	0.009	2.58	1.25–5.31
IV Ascorbic acid	79 (60%)	71 (60%)	0.85	0.95	0.59–1.60
Hydroxychloroquine	98 (78%)	94 (79%)	0.91	1.036	0.56–1.91
Azithromycin	37 (29%)	50 (42%)	0.040	1.74	1.030–2.95
Heparin full dose	72 (55%)	54 (45%)	0.13	0.67	0.41–1.10
Heparin DVT prophylaxis	37 (29%)	48 (40%)	0.06	1.69	1.002–2.38
Convalescent plasma	38 (29%)	27 (23%)	0.24	0.71	0.40–1.25
Remdesivir	5 (4%)	1 (0.9)	0.21	0.20	0.024–1.84
Tocilizumab	10 (7.7%)	16 (13.6%)	0.14	1.86	0.80–4.30
Corticosteroids only	39 (30%)	30 (25%)	0.39	0.80	0.46–1.40
Tocilizumab and steroids	57 (44%)	35 (29%)	0.018	0.53	0.31–0.90

Heparin full dose = anticoagulant dose of unfractionated or low molecular weight heparin. Heparin DVT prophylaxis = 5000 units sc every 8 h or low molecular weight heparin 30–40 mg daily. If patients received both anticoagulant and thrombo-prophylaxis doses during hospitalization they were analyzed in the full-dose group.

**Table 3 biomedicines-11-00845-t003:** Inflammatory and thrombotic markers in COVID-19 patients with and without AKI.

	Non-AKI (n = 130)	AKI (n = 119)	*p*
D-Dimer day 1 (ng/mL)	734 (510,1340)	1169 (470,3680)	0.049
D-Dimer day 2	727 (487,1592)	1380 (471,3680)	0.12
CRP day 1 (mg/L)	108 (52,172)	137 (84,178)	0.26
CRP day 2	105 (46,156)	128 (83,212)	0.026
Ferritin day 1 (ng/mL)	732 (450,1316)	1017 (536,1580)	0.20
Ferritin day 2	839 (524,1665)	906 (567,1756)	0.61

(Biomarker levels are expressed as median with interquartile ranges (IQR)). Abbreviations: CRP = C-reactive protein, AKI = acute kidney injury.

**Table 4 biomedicines-11-00845-t004:** Multivariate analysis of risk factors for AKI.

	Beta	S.E	*p*	OR	95% CI
SCr	0.878	0.221	0.0001	2.406	1.56–3.70
Vasopressor requirement	1.159	0.321	0.0001	3.188	1.69–5.98
Tocilizumab plus CC	−0.827	0.317	0.009	0.437	0.23–0.81
D-Dimer day 1	0.000	0.000	0.008	1.0001	1.000–1.001
CRP day 2	0.004	0.002	0.033	1.004	1.0001–1.009

Abbreviations: 95% CI = 95% confidence interval, CC = corticosteroid therapy, CRP = C-reactive protein, OR = odds ratio, SCr = serum creatinine, S.E. = standard error. For categorical risk factors, the beta coefficient signifies the average change in outcome if that risk factor is present. In the case of continuous risk factors, it signifies the amount the outcome changes for a unit increase in the risk factor’s value. Odds ratios and their 95% confidence intervals (95% CI) are determined by exponentiation of the beta coefficient and its upper and lower CI, respectively.

**Table 5 biomedicines-11-00845-t005:** COVID-19 ICU patients: characteristics of survivors and non-survivors.

	Survivors (n = 87)	Non-Survivors (n = 162)	*p*	OR	95% CI
Age	66 (51, 75)	73 (63, 82)	0.00001		
Race (Caucasian)	54 (62%)	123 (76%)	0.021	1.9	1.09–3.40
BMI	28.9 (24.33)	29 (24, 35)	0.62		
Sex (male)	50 (57%)	98 (60%)	0.64	1.06	0.66–1.92
Diabetes	18 (21%)	58 (36%)	0.014	2.1	1.16–3.90
CHF	8 (9%)	22 (14%)	0.30	1.56	0.66–3.60
CAD	18 (21%)	43 (26%)	0.30	1.38	0.74–2.6
COPD	22 (25%)	38 (23%)	0.74	0.9	0.50–1.6
CKD	8 (7%)	20 (12%)	0.18	1.90	0.73–4.90
HTN	39 (45%)	96 (59%)	0.029	1.80	1.06–3.0
AKI	31 (35%)	88 (54%)	0.005	2.14	1.25–3.70
Cirrhosis	0 (0%)	4 (2%)	0.3	0.64	0.58–0.7
Malignancy	7 (8%)	20 (12%)	0.39	1.61	0.65–3.97
CVA	9 (10%)	18 (11%)	0.85	1.1	0.46–2.52
Mechanical ventilation	43 (49%)	130 (80%)	0.00001	4.10	2.3–7.30
Dialysis *	3 (3.4%)	20 (12%)	0.022	3.94	1.13–13.6
Neutrophiles × 10^9^/L	8 (4.5, 13)	7.4 (5, 11.7)	0.67		
Lymphocytes × 10^9^/L	0.9 (0.6, 1.6)	0.7 (0.5, 1.1)	0.010		
Neutrophile/lymphocyte	7.6 (4.3, 14)	10 (6, 18.4)	0.03		
SCr µmoles/L)	95.4 (55.2, 141.4)	106 (72.5, 155)	0.12		
Plts × 10^9^/L	262 (195, 326)	226 (151, 279)	0.019		
Tbili (µmoles/L)	8.5 (5.1, 12)	10.3 (6.8, 16.4)	0.012		
SOFA	4 (2, 6)	4 (3, 7)	0.095		
PaO2/FIO2	231 (118, 310)	191 (79, 286)	0.051		
Pa02 KPa	9.7 (7.8, 13)	9.2 (7.3, 11.4)	0.14		
FI02	0.36 (0.29, 0.96)	0.40 (0.21, 1.0)	0.12		

Abbreviations: AKI = acute kidney injury, BMI = body mass index, CAD = coronary artery disease, CHF = congestive heart failure, CI = confidence interval, CKD = chronic kidney disease, COPD = chronic obstructive pulmonary disease, HD = hemodialysis, HTN = hypertension, OR = odds ratio, PaO2/FiO2 = ratio of partial pressure of oxygen to inspired concentration of oxygen, Plts = platelets, SCr = serum creatinine, SOFA = Sepsis-Related Organ Failure Assessment, TBili = total bilirubin. * Requirement for dialysis that developed within the first 7 days after hospital admission.

**Table 6 biomedicines-11-00845-t006:** Risk factors for mortality among COVID-19 ICU patients with AKI.

	Survivor (n = 31)	Non-Survivor (n = 88)	*p*	OR	95% CI
Age	69 (63, 77)	75 (65, 82)	0.1		
Race (Caucasian)	19 (63%)	69 (78%)	0.10	2.10	0.85–5.1
BMI	26 (24.32)	29 (25, 36)	0.10		
Sex (male)	18 (60%)	58 (66%)	0.50	1.20	0.52–3.0
Diabetes	11 (36%)	30 (34%)	0.80	0.89	0.38–2.10
CHF	5 (17%)	12 (14%)	0.70	0.8	0.25–2.50
CAD	9 (30%)	20 (23%)	0.40	0.68	0.27–1.70
COPD	33 (25%)	27 (33%)	0.67	0.88	0.49–1.58
CKD	6 (20%)	9 (10%)	0.16	0.6	0.23–1.50
HTN	18 (60%)	51 (58%)	0.84	0.9	0.39–2.10
Cirrhosis	0 (0%)	2 (2.3%)	1.0	0,75	0.66–0.82
Malignancy	4 (13%	9 (10, 2%)	0.74	0.77	0.22–2.7
CVA	6 (19%)	8 (9%)	0.13	0.42	0.13–1.32
Mechanical ventilation	16 (53%)	77 (87%)	0.00001	6.1	2.30–16.0
Dialysis *	3 (9.7%)	20 (22.7%)	0.18	2.78	0.75–9.98
Neutrophiles × 10^9^/L	7.7 (4.6, 14)	7.5 (5.3, 13)	0.49		
Lymphocytes × 10^9^/L	1.0 (0.67, 1.9)	0.7 (0.5, 1.)	0.004		
Neutrophile/lymphocyte	7.30 (3.8, 12.6)	10.5 (6, 19)	0.009		
SCr µmole/L	159 (115, 256.4)	132.6 (88.4, 177.6)	0.04		
Plts × 10^9^/L	281 (212, 326)	223 (141, 277)	0.010		
Tbili µmole/L	6.8 (5.1, 13.7)	8.5 (6.8, 15.4)	0.12		
SOFA admit	4.5 (3, 8.5.5)	5 (3, 8)	0.80		
PaO2/FIO2	227 (122, 307)	200 (74, 313)	0.25		
Pa02 KPa	9.8 (8, 12)	9 (7.2, 11.3)	0.10		
FI02	0.34 (0.25, 0.96)	0.42 (0.21, 1)	0.40		

Abbreviations: AKI = acute kidney injury, BMI = body mass index, CAD = coronary artery disease, CHF = congestive heart failure, CI = confidence interval, CKD = chronic kidney disease, COPD = chronic obstructive pulmonary disease, HD = hemodialysis, HTN = hypertension, OR = odds ratio, PaO2/FiO2 = ratio of partial pressure of oxygen to inspired concentration of oxygen, Plts = platelets, SCr = serum creatinine, SOFA = Sepsis-Related Organ Failure Assessment, TBili = total bilirubin. * Requirement for dialysis that developed within the first 7 days after hospital admission.

**Table 7 biomedicines-11-00845-t007:** Risk factors for mortality among COVID-19 ICU patients without AKI.

	Survivor (n = 56)	Non-Survivor (n = 74)	*p*	OR	95% CI
Age	62 (44, 68)	70 (63, 80)	0.00001		
Race (Caucasian)	34 (60%)	54 (73%)	0.14	1.70	0.81–3.70
BMI	29 (26.34)	29 (24, 33)	0.52		
Sex (male)	31 (55%)	40 (54%)	0.88	0.95	0.47–1.90
Diabetes	7 (12%)	28 (38%)	0.001	4.3	1.70–11.0
CHF	3 (5.4%)	10 (13%)	0.13	2.8	0.72–10.5
CAD	9 (16%)	23 (31%)	0.049	2.40	1.0–5.70
COPD	13 (23%)	20 (27%)	0.6	1.22	0.54–2.70
CKD	0 (0%)	11 (15%)	0.003	9.9	1.20–79.0
HTN	21 (37%)	45 (61%)	0.005	2.79	1.36–5.70
Cirrhosis	0 (0%)	2 (2.7%)	0.50	0.56	0.48–0.65
Malignancy	3 (5.4%)	11 (15%)	0.095	3.1	0.82–11.6
CVA	3 (5.4%)	10 (13.5%)	0.15	2.76	0.72–10.4
Mechanical ventilation	26 (47%)	53 (71%)	0.005	2.90	1.4–6.0
Neutrophiles × 10^9^/L	8.6 (4.6, 15.0)	6.6 (4.4, 10.5)	0.22		
Lymphocytes × 10^9^/L	0.9 (0.5, 1.40)	0.7 (0.5, 1.2)	0.35		
Neutrophile/lymphocyte	8 (5, 15.3)	9 (6, 17)	0.64		
SCr (µmoles/L)	79.5 (53, 97)	88.4 (62, 123.8)	0.048		
Plts × 10^9^/L	248 (176, 330)	231 (164, 297)	0.30		
Tbili µmoles/L	49.2 (5.1, 12)	10.2 (7.7, 13.6)	0.03		
SOFA admit	3 (2.0.5.0)	4 (3, 6)	0.17		
PaO2/FIO2	230 (116, 309)	190 (98, 261)	0.12		
Pa02 KPa	9.7 (7.3, 12.7)	9.2 (7.4, 11.4)	0.53		
FI02	0.36 (0.21, 0.96)	0.40 (0.26, 1)	0.29		

Abbreviations: AKI = acute kidney injury, BMI = body mass index, CAD = coronary artery disease, CHF = congestive heart failure, CI = confidence interval, CKD = chronic kidney disease, COPD = chronic obstructive pulmonary disease, HD = hemodialysis, HTN = hypertension, OR = odds ratio, PaO2/FiO2 = ratio of partial pressure of oxygen to inspired concentration of oxygen, Plts = platelets, SCr = serum creatinine, SOFA = Sepsis-Related Organ Failure Assessment, TBili = total bilirubin.

**Table 8 biomedicines-11-00845-t008:** Pharmacologic interventions in COVID-19 ICU survivors and non-survivors.

	Non-Survivor (n = 162)	Survivor (n = 87)	*p*	OR	95% CI
Vasopressors	119 (73%)	33 (38%)	0.00001	4.50	2.60–7.90
IV Ascorbic acid	99 (59%)	51 (59%)	0.70	1.1	0.66–1.89
Hydroxychloroquine	124 (78%)	68 (80%)	0.71	0.88	0.46–1.69
Azithromycin	63 (40%)	24 (28%)	0.071	1.69	0.97–3.0
Heparin full dose	79 (49%)	47 (54%)	0.51	0.80	0.48–1.36
Heparin DVT prophylaxis	54 (33%)	31 (36%)	0.8	0.90	0.52–1.60
Convalescent plasma	42 (26%)	23 (26%)	0.93	0.97	0.54–1.76
Remdesivir	6 (3%)	0 (0)	0.094	0.64	0.680.70
Tocilizumab	18 (11%)	8 (9.2%)	0.80	1.23	0.51–2.98
Corticosteroids only	44 (27%)	25 (28%)	0.47	0.80	0.48–1.40
Tocilizumab and steroids	55 (34%)	37 (42%)	0.18	0.69	0.40–1.86

Heparin full dose = anticoagulant dose of unfractionated or low molecular weight heparin. Heparin DVT prophylaxis = 5000 units sc every 8 h or low molecular weight heparin 30–40 mg daily. If patients received both anticoagulant and thrombo-prophylaxis doses during hospitalization they were analyzed in the full-dose group.

**Table 9 biomedicines-11-00845-t009:** Pharmacologic interventions in COVID-19 ICU survivors and non-survivors with AKI.

	Survivors (n = 31)	Non-Survivors (n = 88)	*p*	OR	95% CI
Vasopressors	14 (48%)	72 (82%)	0.00001	4.8	2.0–12.0
IV Ascorbic acid	21 (68%)	50 (57%)	0.28	0.62	0.26–1.48
Hydroxychloroquine	22 (71%)	72 (82%)	0.21	1.84	0.71–4.70
Azithromycin	11 (35%)	39 (44%)	0.39	1.44	0.62–3.37
Heparin full dose	18 (58%)	36 (41%)	0.09	0.50	0.22–1.15
Heparin DVT prophylaxis	11 (36%)	37 (41%)	0.090	0.50	0.22–3.10
Convalescent plasma	7 (23%)	20 (23%)	0.94	0.96	0.36–2.68
Remdesivir	0 (0%)	1 (0.9)	0.5	0.73	0.66–0.82
Tocilizumab	5 (16%)	11 (12%)	0.56	0.76	0.3–1.9
Corticosteroids only	9 (29%)	21 (23%)	0.57	1.12	0.52–2.4
Tocilizumab and steroids	12 (39%)	23 (26%)	0.18	0.56	0.23–1.33

Heparin full dose = anticoagulant dose of unfractionated or low molecular weight heparin. Heparin DVT prophylaxis = 5000 units sc every 8 h or low molecular weight heparin 30–40 mg daily. If patients received both anticoagulant and thrombo-prophylaxis doses during hospitalization they were analyzed in the full-dose group.

**Table 10 biomedicines-11-00845-t010:** Pharmacologic interventions in COVID-19 ICU survivors and non-survivors without AKI.

	Survivors (n = 56)	Non-Survivors (n = 74)	*p*	OR	95% CI
Vasopressors	18 (32%)	47 (63%)	0.00001	3.67	1.76–7.65
IV Ascorbic acid	30 (54%)	49 (66%)	0.14	1.70	0.83–3.50
Hydroxychloroquine	46 (85%)	52 (73%)	0.11	0.47	0.19–1.20
Azithromycin	13 (24%)	24 (34%)	0.21	1.65	0.75–3.64
Heparin full dose	29 (52%)	43 (58%)	0.47	1.29	0.64–2.60
Heparin DVT prophylaxis	20 (36%)	17 (44%)	0.10	0.53	0.25–1.20
Convalescent plasma	16 (28%)	22 (30%)	0.86	1.06	0.49–2.27
Remdesivir	0 (0%)	5 (6.8)	0.070	0.54	0.46–0.64
Tocilizumab	3 (5%)	7 (9%)	0.51	1.86	0.45–7.50
Corticosteroids only	16 (29%)	23 (31%)	0.75	1.12	0.69–1.66
Tocilizumab and steroids	25 (44%)	32 (43%)	0.87	0.94	0.47–1.90

Heparin full dose = anticoagulant dose of unfractionated or low molecular weight heparin. Heparin DVT prophylaxis = 5000 units sc every 8 h or low molecular weight heparin 30–40 mg daily. If patients received both anticoagulant and thrombo-prophylaxis doses during hospitalization they were analyzed in the full-dose group.

**Table 11 biomedicines-11-00845-t011:** Inflammatory and thrombotic markers in COVID-19 ICU survivors and non-survivors.

All ICU Patients	Survivors (n = 87)	Non-Survivors (n = 162)	*p*
D-Dimer day 1 (ng/mL)	636 (337,1502)	852 (520,2309)	0.069
D-Dimer day 2	1166 (548,3555)	691 (436,1743)	0.089
CRP day 1 (mg/L)	134 (90,204)	124 (61,175)	0.41
CRP day 2	123 (47,171)	115 (81,184)	0.38
Ferritin day 1 (ng/mL)	995 (352,1571)	790 (400,1460)	0.25
Ferritin day 2	987 (675,1888)	822 (462,1478)	0.12
**ICU Patients without AKI**	**Survivors** **(n = 56)**	**Non-Survivors** **(n = 74)**	** *p* **
D-Dimer day 1	629 (330,1033)	799 (632,1432)	0.092
D-Dimer day 2	900 (548,1814)	646 (481,839)	0.12
CRP day 1	130 (58,232)	103 (51,162)	0.35
CRP day 2	97 (38,157)	105 (73,152)	0.59
Ferritin day 1	735 (471,1347)	699 (416,1338)	0.80
Ferritin day 2	987 (691,1972)	748 (446,1374)	0.19
**ICU Patients with AKI**	**Survivors** **(n= 31)**	**Non-Survivors** **(n = 88)**	** *p* **
D-Dimer day 1	644 (335,3680)	1219 (500,3083)	0.71
D-Dimer day 2	2390 (543,3680)	1182 (419,3436)	0.28
CRP day 1	137 (93,169)	140 (21,185)	0.86
CRP day 2	140 (80,204)	122 (83,223)	0.89
Ferritin day 1	1390 (1003,2318)	923 (320,1542)	0.070
Ferritin day 2	1139 (657,2130)	993 (447,1590)	0.30

Abbreviations: CRP = C-reactive protein.

**Table 12 biomedicines-11-00845-t012:** Multivariate Cox proportional hazards analysis of risk factors for mortality.

All ICU Patients	B	SE	*p*	HR	95% CI
Age	0.028	0.006	0.00001	1.028	1.016–1041
**ICU Patients with AKI**					
SCr	−2.32	0.092	0.01	0.79	0.66–0.95
**ICU Patients without AKI**					
Age	0.043	0.01	0.00001	1.044	1.024–1.065
CKD	0.97	0.34	0.004	2.65	1.37–5.1

Abbreviations: 95% CI = 95% confidence interval, CKD = chronic kidney disease, HR = hazards ratio, SCr = serum creatinine, S.E. = standard error.

## Data Availability

The data used in this study are protected under HIPPA and were not intended or allowed to be shared publicly, so due to the sensitive nature of the research supporting data are not available.

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
