# Peer review of "Acute Kidney Injury Associated with Severe SARS-CoV-2 Infection: Risk Factors for Morbidity and Mortality and a Potential Benefit of Combined Therapy with Tocilizumab and Corticosteroids"

_biomedicines, 2023, doi:10.3390/biomedicines11030845_

Round 1

Reviewer 1 Report

The manuscript is interesting but some points must be improved. In particular: 

Introduction: It must be pointed out that COVID-19 is the disease caused by SARS-CoV-2 infection.

Lines 55-59: The complexity of COVID-19 disease deserves to be highlighted since it has been reported that SARS-CoV2 infection can lead to respiratory but also non-respiratory disease as recently review (PMID: 35943095, 35114008, 36072173).This is an important point to add since it can further highlight the important results found by the authors.

Line 74: Authors must clarify which characteristics (symptoms, signs....etc) the patients must have to be classified as "severe" COVID-19 disease

Tables format: authors should use the table format present in the Biomedicines template 

An accurate revision of punctuation and headings  is recommended

Author Response

Introduction:

It must be pointed out that COVID-19 is the disease caused by SARS-CoV-2 infection.

We do so now in the first sentence of the Introduction. The sentence now reads: "Severe Covid-19 disease, as caused by SARS-CoV2 infection, results in a wide variety..."

(line 42)

Lines 55-59:

The complexity of COVID-19 disease deserves to be highlighted since it has been reported that SARS-CoV2 infection can lead to respiratory but also non-respiratory disease as recently review (PMID: 35943095, 35114008, 36072173). This is an important point to add since it can further highlight the important results found by the authors.

We agree that this is an important point.  We have added the following new text to address this issue:

"Although Covid-19 disease primarily impacts the respiratory tract, the enzyme angiotensin converting enzyme-2 (ACE-2), the primary receptor for SARS-Cov-2, is expressed by the endothelium of almost all extra-pulmonary organs, including the reproductive tract.  Interaction of SARS-CoV-2 with the ACE-2 receptor leads to a prothrombotic hyper-inflammatory state as well as maladaptive hyper-activation of the innate immune system resulting in cytokine storm.  In this respect, Covid-19 disease can be viewed as a multisystem disease, with both acute and chronic multiorgan dysfunction, including myocarditis, myocardial infarction, macrophage activation syndrome, arterial and venous  thrombosis, encephalitis, stroke, thyroiditis, adrenal insufficiency, hepatitis, hepatic failure, preeclampsia, and sterility."

(lines 55-64)

Line 74: 

Authors must clarify which characteristics (symptoms,signs....etc) the patients must have to be classified as "severe"COVID-19 disease

For the purposes of this study, "severe COVID-19 disease" was defined as direct admission or transfer to an ICU.  However, all patients fulfilled the following criteria for severe disease:

"At the time of ICU entry, all patients manifested severe difficulty in breathing as evidenced by one or more of the following: respiratory rate greater than 30 breaths per minute, blood oxygen saturation of 93% or less on room air, PAO2/FIO2 ratio less than 300, and presence of lung infiltrates in more than half of the lung fields."

(lines 92-95)

Tables format:

authors should use the table format present in the Biomedicines template

As per communication with the Editor, given the short deadline for revision, proper formatting of Tables has been deferred until after formal acceptance.

Reviewer 2 Report

An interesting retrospective study with some shortcomings.

Minor:

Some textual errors, for example in the abstract "elevated CRP on day 2 (p=033)"?, missing dot at the end of the conclusion, etc.

Major:

There are no declared analytical methods for laboratory analyzes (including biomarkers) with basic analytical characteristics. The statistical program with which the results were processed is not specified. Among the stated weaknesses of the study (no data on urine output, etc. ) are also many missing results, especially for biomarkers, so it's difficult to make the conclusion about significant risk factors for AKI from the remaining results despite "replaced values using the expectation maximization algorithm".

Author Response

Minor:

Some textual errors, for example in the abstract "elevated CRP on day 2 (p=033)"?, missing dot at the end of the conclusion, etc.

We thank the Reviewer for his/her careful eye in picking up these inadvertent errors.  They have been corrected.

Major:

There are no declared analytical methods for laboratory analyzes (including biomarkers) with basic analytical characteristics.

We have added the following text:

"All standard laboratory studies were extracted from the electronic medical record and were performed according to standardized laboratory practices.  D-dimer levels were obtained employing a latex agglutination photo optical assay from a sample of citrated whole blood.  CRP levels were obtained from serum samples and analyzed incorporating a laser-nephelometric method."

(lines 104-108)

The statistical program with which the results were processed is not specified.

We have added the requested information:

"All statistical analyses were performed with IBMâ SPSSâ software."

(line 132)

Among the stated weaknesses of the study (no data on urine output, etc. ) are also many missing results, especially for biomarkers, so it's difficult to make the conclusion about significant risk factors for AKI from the remaining results despite "replaced values using the expectation maximization algorithm."

We agree with the reviewer and have therefore inserted the following additional weakness into our Discussion:

"Sixth, there were many missing data points for the biomarkers D-Dimer, CRP, and ferritin, thus making it difficult to draw robust conclusions regarding their role as risk factors in the development of AKI.  As a result, we had to rely on imputed data.  Employing the imputed values into the multivariate analysis should therefore be considered exploratory.  Nonetheless, removing the imputed biomarker data from the multivariate analysis did not alter the remaining risk factors.  Further studies are needed to elucidate the role of biomarkers as predictors of AKI in patients with Covid-19 disease."

(lines 371-378)

Reviewer 3 Report

The Authors Jose Iglesias et al., in the manuscript entitled Acute Kidney Injury Associated with Severe SARS-CoV-2 Infection: Risk Factors for Morbidity and Mortality and a Potential Benefit of Anti-Inflammatory therapy,  show a retrospective observational study to analyze risk factors associated with AKI.

The manuscript needs to be updated with other more recent references on the  topics AKI and micro coagulopathies in COVID patients. There are several punctuation and data errors in some references. The manuscript could be improved. I invite the authors to review the manuscript in general and correct the major and minor revisions.

Important: The manuscript needs thorough revision as the graphs of the results are missing. The Tables do not follow the journal's style and guidelines, so they need to be remodeled. I look forward to reading the revised manuscript.

 Major revisions:

11.    Graphs of statistical analysis, univariate and multivariate, are missing. Please add.

22.     The name of the software used for the statistical analysis should also be given. Please add.

3.3.   Line 186: tocilizumab was taken by a low percentage (table 2) of AKI and non-AKI subjects; the conclusion seems to be too forced. Please reshape the sentence in terms of probability.

44.  Line 165-167: what was the selection criterion for the different drugs? why do some patients have hydroxychloroquine and others azithromycin? the rationale? the time of administration? the modality? How many days? If it is not possible, the authors must consider removing the drugs' data from the results or show a graph of multivariate analysis.

Minor revision

1.      A potential effect of anti-inflammatories is illustrated in the title, but very little of the topic is covered in the discussion. The authors need to adjust the title or extend a lot the discussion when explaining the possible rationale for using anti-inflammatories.

2.      Line 27: Please add  “ Sequential Organ Failure Assessment” (SOFA)

3.      Line 24, please add the “intensive care unit” (ICU)

4.      Line 28: CRP (C-reactive protein)

5.      Lines  38,40,43: re- write Covid in COVID everywhere or vice versa (lines 24,35, eg.)

6.      Line 40: remove the number and add other keywords like D-Dimer, NSAID, and coagulopathy.

7.      Lines 42-45: Do The authors refer to the variety of renal injuries in hospitalized patients? Please add or specify it.

8.      Lines 51: please insert the period after the parenthesis everywhere.

9.      Lines 53: C19-ARDS instead of CARDS, everywhere or cite then references where this term was used in the first time.

10.   Line 54: the reference does not show this data. Maybe it is wrong; cite the right reference and indicate the period and virus variant in which this data is evaluated.

11.   Line 57 is not around 7 days, but from 7 to 14 days; please be careful when reporting the data of other papers and indicate the period and the viral variant. Check again all data reported in the all the manuscript.

12.   Line 59, please add the mechanism of coagulation also:

a.      Gerlach, Joachim et al. “The immune paradox of SARS-CoV-2: Lymphocytopenia and autoimmunity evoking features in COVID-19 and possible treatment modalities.” Reviews in medical virology, e2423. 2 Feb. 2023, doi:10.1002/rmv.2423

b.      Asakura, Hidesaku, and Haruhiko Ogawa. “COVID-19-associated coagulopathy and disseminated intravascular coagulation.” International journal of hematology vol. 113,1 (2021): 45-57. doi:10.1007/s12185-020-03029-y

13.   Line 163 is missing the indication of where to see the graph showing this result.

14.   Line 176: we looked sound too friendly better “we study”.

15.   Line 238 : Insert the other abbreviations

16.   Line 210: In Table 11, it appears that d-dimer values are high especially in non-survivors. Remodel this sentence

17.   Line 239, table 4: Beta?? indicate in legend its meaning

18.   Line 335: add thrombotic microangiopathies (TMA)

19.   Line 367: modulate the sentence because the cases in tables are few treated with tolicuzamb compared to the majority

20.   Line 371: Argue more by adding more references to the aspect of thromboembolisms in Covid/COVID  patients

21.   Line 381: add the reference

22.   Line 387: Lately, the bacteriophagic behavior of SARS-CoV-2 and the presence of microbiome toxins have also been observed. This could be another explanation, along with the many so far seen. The authors should keep these more recent studies in mind as well.

a.      Brogna, Carlo et al. “Toxin-like peptides in plasma, urine and faecal samples from COVID-19 patients.” F1000Research vol. 10 550. 8 Jul. 2021, doi:10.12688/f1000research.54306.2

b.      Petrillo, Mauro et al. “Increase of SARS-CoV-2 RNA load in faecal samples prompts for rethinking of SARS-CoV-2 biology and COVID-19 epidemiology.” F1000Research vol. 10 370. 11 May. 2021, doi:10.12688/f1000research.52540.3

c.  https://www.nature.com/articles/s41467-023-35787-8 Cheney, A.M., Costello, S.M., Pinkham, N.V. et al. Gut microbiome dysbiosis drives metabolic dysfunction in Familial dysautonomia. Nat Commun 14, 218 (2023). https://doi.org/10.1038/s41467-023-35787-8

Author Response

Major revisions:

  1. Graphs of statistical analysis, univariate and multivariate, are missing. Please add.

We have replaced both multivariate tables with Forrest plots.  We have also added a Forrest plot for the univariate analysis of pharmacologic agents.  We feel, however, that to include Forrest plots for the other univariate analyses would be unnecessarily busy and distracting.  The tables and Forrest plots contain identical information, their only difference being in format.  We will defer on this issue to editorial discretion.

  1. The name of the software used for the s22.

The name of the software used for the statistical analysis should also be given. Please add.

We have added the requested information:

"All statistical analyses were performed with IBMâ SPSSâ software."

(line 132)

3.3. Line 186: tocilizumab was taken by a low percentage (table 2) of AKI and non-AKI subjects; the conclusion seems to be too forced. Please reshape the sentence in terms of probability.

The Reviewer is correct that Tocilizumab as a monotherapy was employed in only a minority of patients.  However, Toculizimab was also administered as dual therapy in combination with CC, so that a total of 118 patients (47%) received Toculizimab, either as monotherapy (26) or dual therapy with CC (92).  We acknowledge that there may be confusion in the wording of the text.  We have therefore added the following sentence:

"The use of either Tocilizumab alone or CC alone as monotherapy was not associated with a reduced risk of AKI.  Only the combination predicted a lower risk of AKI."

(line 204-206)

  1. Line 165-167: what was the selection criterion for the different drugs? why do some patients have hydroxychloroquine and others azithromycin? the rationale? the time of administration? the modality? How many days? If it is not possible, the authors must consider removing the drugs' data from the results or show a graph of multivariate analysis.

This being a retrospective analysis, the investigators had no control over the rationale, route of administration, or timing of therapeutics.  Information on timing was available only for corticosteroids and Tocilizumab.  We have added the following additional text:

"In general, information on the timing, rationale, and route of administration of agents was available only for CC and Tocilizumab and is therefore not part of the analysis."

(lines 191-193)

"There was no difference in the time of initiation of CC among patients with and without AKI (4 days IQR (1,14) versus 3 days (IQR 1,8), p=0.56).  Similarly, there was no difference in the time of initiation of Tocilizumab among patients with and without AKI (4 days IQR (2,6) versus 3 days IQR (1.5,7), p=0.96)."

(lines 206-209)

Minor revision

  1. A potential effect of anti-inflammatories is illustrated in the title, but very little of the topic is covered in the discussion. The authors need to adjust the title or extend a lot the discussion when explaining the possible rationale for using anti-inflammatories.

The mention of anti-inflammatories in the title is based on the multivariate assessment of risk factors for the development of AKI, in which the combined use of Tocilizumab and CC (but not either alone) was associated with a reduced risk of AKI.  An alternate more specific title would be to replace "Anti-Inflammatory Therapy" in the title with "Combined Therapy with Tocilizumab and Corticosteroids".  We defer on this point to editorial discretion.

  1. Line 27: Please add

“ Sequential Organ Failure Assessment” (SOFA)

This has been added.

  1. Line 24, please add the “intensive care unit” (ICU)

This has been added.

  1. Line 28: CRP (C-reactive protein)

This has been added.

  1. Lines 38,40,43: re- write Covid in COVID everywhere or vice versa (lines 24,35, eg.)

We have changed all COVID to Covid in the text.

  1. Line 40: remove the number and add other keywords like D-Dimer, NSAID, and coagulopathy.

We have modified the list of key words

  1. Lines 42-45: Do The authors refer to the variety of renal injuries in hospitalized patients? Please add or specify it.

We are unsure what the Reviewer is requesting here.  The cited lines specify the variety of renal injuries that have been found in patients with Covid-19 infection.  Without a renal biopsy, we cannot determine which of these pathologies was present in the patients of our study.  Indeed, as we discuss in the 3rd paragraph of the Discussion, it remains unclear whether Covid-19-associated AKI is itself a separate clinical entity.  The focus of our study was on AKI, in general, without distinguishing the underlying pathology.

  1. Lines 51: please insert the period after the parenthesis everywhere.

The requested change will be reflected in the final accepted version of our manuscript.

  1. Lines 53: C19-ARDS instead of CARDS, everywhere or cite then references where this term was used in the first time.

We have made the requested change.

  1. Line 54: the reference does not show this data. Maybe it is wrong; cite the right reference and indicate the period and virus variant in which this data is evaluated.

We believe that the cited reference is correct.  The requested information has been added.

"The prevalence of AKI in non-survivors of Covid-19 with C19-ARDS infected with the Wuhan strain of the virus during the first wave of the pandemic..."

(lines 52-53)

  1. Line 57 is not around 7 days, but from 7 to 14 days; please be careful when reporting the data of other papers and indicate the period and the viral variant. Check again all data reported in the all the manuscript.

We have made the correction.

(line 67)

  1. Line 59, please add the mechanism of coagulation also:

a.Gerlach, Joachim et al. “The immune paradox of SARS-CoV-2:Lymphocytopenia and autoimmunity evoking features in COVID-19 and possible treatment modalities.”

Reviews in medical virology, e2423. 2 Feb. 2023,doi:10.1002/rmv.2423

b.Asakura, Hidesaku, and Haruhiko Ogawa. “COVID-19-associatedcoagulopathy and disseminated intravascular coagulation.”

International journal of hematology

vol. 113,1 (2021):45-57. doi:10.1007/s12185-020-03029-y

We have added the following sentence.

"An imbalance among pro- and anti-anticoagulant factors, in part a result of hyper-inflammation and cytokine storm, leads to a prothrombotic state characterized by disseminated intravascular coagulation, macro- and microvascular thrombotic events, and multi-organ failure."

(lines 70-73)

  1. Line 163 is missing the indication of where to see the graph showing this result.

This is now indicated.

  1. Line 176: we looked sound too friendly better “we study”.

We have made the requested change.

  1. Line 238 : Insert the other abbreviations

The only missing abbreviation is AKI, and it has been inserted.

  1. Line 210: In Table 11, it appears that d-dimer values are high especially in non-survivors. Remodel this sentence

The p values are not significant for any of the comparisons with respect to D-Dimer in Table 11.  Indeed, the relationship between D-Dimer levels for survivors versus non-survivors is not always in a consistent direction.  It would be incorrect and an over-interpretation of our data to discuss any appearances for these insignificant differences. 

17.Line 239, table 4: Beta?? indicate in legend its meaning

A footnote now appears explaining the beta coefficient.

"For categorical risk factors, the β coefficient signifies the average change in outcome if that risk factor is present.  In the case of continuous risk factors, it signifies the amount the outcome changes for a unit increase in the risk factor's value.  Odds ratios and their 95% confidence intervals (95% CI) are determined by exponentiation of the beta coefficient and its upper and lower CI, respectively."

  1. Line 335: add thrombotic microangiopathies(TMA)

TMA has already been defined in the first sentence of the Introduction.

  1. Line 367: modulate the sentence because the cases in tables are few treated with tolicuzamb compared to the majority

We have already addressed this point and made changes to the text.  Please refer to our response to this Reviewer's 3rd major revision above.

  1. Line 371:

Argue more by adding more references to the aspect of thromboembolisms in Covid/COVID patients

We have added the following sentence:

"Moreover a systematic review has shown the risk of both venous and arterial thromboembolism to be statistically significantly higher in those with Covid-19 infection relative to those without (Malas et al, 2020)."

(lines 336-338)

  1. Line 381: add the reference

The reference has been added.

  1. Line 387: Lately, the bacteriophagic behavior of SARS-CoV-2 and the presence of microbiome toxins have also been observed. This could be another explanation, along with the many so far seen. The authors should keep these more recent studies in mind as well.
  2. Brogna, Carlo et al. “Toxin-like peptides in plasma, urine and faecal samples fromCOVID-19 patients.”

F1000Research

vol.10 550. 8 Jul. 2021,doi:10.12688/f1000research.54306.2

  1. Petrillo, Mauro et al. “Increase of SARS-CoV-2 RNA load in faecal samples prompts for rethinking of SARS-CoV-2biology and COVID-19epidemiology.”

F1000Research

vol. 10370. 11 May. 2021,doi:10.12688/f1000research.52540.3

  1. https://www.nature.com/articles/s41467-023-35787-8 Cheney, A.M., Costello,S.M., Pinkham, N.V. et al. Gut microbiome dysbiosis drives metabolic dysfunction in Familial dysautonomia. Nat Commun 14, 218 (2023).https://doi.org/10.1038/s41467-023-35787-8

We appreciate the Reviewer's calling these papers to our attention.  While of interest, we feel that discussion of the role of the microbiome lies outside the scope of our manuscript.

Reviewer 4 Report

The authors study the risk factors for morbidity and mortality of acute kidney injury associated with severe SARS-CoV-2 Infection. The authors revealed the risk factors of therapeutic and pharmacologic interventions, inflammatory and coagulopathic markers to the development of AKI, and mortality using univariate and multivariable analysis. These studies guide prevention and therapy in AKI development associated with Covid-19. Before Biomedicine accepts this manuscript, the following concerns should be addressed.

Major comments:

  1. Besides the characteristics, past medical history should also be considered, especially for the survival evaluation.  
  2. The conclusion cannot fully include or represent the context. Please include the essential findings about risk factors for AKI in the conclusion section.
  3. Line 104 to 109, the definition of AKI needs to be more comprehensive. Explanation and support information are required to justify the arbitrarily assigned 1 mg/dl value. 
  4. Line 136, please define severe Covid-19 Infection in this study. 
  5. Please include more description and conclusion in Lines 177-178 for table 3.
  6. Please indicate the reason for cox proportional hazards analysis and forward variable selection.

Minor comments: 

  1. Please spell out SOFA and CPR in the abstract. 
  2. Please remove the P, OR, and 95% CI values from the abstract. 
  3. Line 153, please include a comma between the number 3 and IQR. 
  4. Please add a note to table 3 indicating the meaning of numbers in parentheses. 
  5. Provide a rationale in the text that why coagulopathic markers are an essential risk factor that was necessary to summarize in table 3. 

Author Response

Major comments:

  1. Besides the characteristics, past medical history should also be considered, especially for the survival evaluation.

We agree with the Reviewer regarding the importance of past medical history.  Elements of the past medical history, as extracted from the medical record based on ICD-10 coding or from the patient's history and physical, are included as comorbidities in Table 1 for evaluation of AKI and in Tables 5, 6, and 7 for evaluation of survival.  None of these comorbidities emerged as a significant predictor of AKI or survival.  We speculate on the reason for this in the Discussion (lines 283-286).  We have also added the following additional co-morbidities to Tables 1, 5, 6, and 7: malignancy, cirrhosis, and cerebrovascular accident.

  1. The conclusion cannot fully include or represent the context. Please include the essential findings about risk factors for AKI in the conclusion section.

We have modified the Conclusion:

"The current study confirms the high rates of AKI and mortality among Covid-19 patients admitted to an ICU.  As seen with other causes of AKI, multivariate analysis demonstrated that an elevated SCr on admission and the need for vasopressors were independent predictors of the development of with AKI in patients with severe Covid-19.  In addition, elevations of CRP and D-Dimer, biomarkers for inflammation and thrombosis, respectively, were also associated with an increased risk for AKI, albeit the effect was of small magnitude.  Notably, patients who received both CC and Tocilizumab, but not either alone, were less likely to develop AKI.   As has been shown for the development of C-19 ARDS in severely ill Covid-19 patients, our data suggest a potential role for inflammation in the development of Covid-19 associated AKI.  One should consider the possibility that early administration of anti-inflammatory agents, just as one does in the management of C-19 ARDS, may improve clinical outcomes in patients with AKI."

(lines 385-396)

  1. Line 104 to 109, the definition of AKI needs to be more comprehensive. Explanation and support information are required to justify the arbitrarily assigned 1 mg/dl value.

We apologize to the Reviewer.  The arbitrary assignment of a value of 1 mg/dL was a leftover from a very early version of the manuscript.  We used the KDIGO definition of AKI.  This is now clarified:

"AKI was defined according to Kidney Disease: Improving Global Outcomes (KDIGO) criteria, namely, an increase in serum creatinine value (SCr) by ≥ 0.3 mg/dL (≥ 26.5  μMol/L) within 48 hours.  Urine output was not considered.  SCr on admission was used to assess the change in SCr."

(lines 126-129)

  1. Line 136, please define severe Covid-19 Infection in this study.

For the purposes of this study, "severe COVID-19 disease" was defined as direct admission or transfer to an ICU.  However, all patients fulfilled the following criteria for severe disease:

"At the time of ICU entry, all patients manifested severe difficulty in breathing as evidenced by one or more of the following: respiratory rate greater than 30 breaths per minute, blood oxygen saturation of 93% or less on room air, PAO2/FIO2 ratio less than 300, and presence of lung infiltrates in more than half of the lung fields."

(lines 92-95)

  1. Please include more description and conclusion in Lines 177-178 for table 3.

We have expanded the text as follows:

"Finally, since endothelial dysfunction, thrombosis, and inflammation are common features of COVID-19 disease, we examined the levels of inflammatory and coagulopathic markers in these patients.  These data were available from the electronic record, as it was common practice during the first wave of the pandemic to obtain serial inflammatory and thrombotic biomarkers in patients with Covid-19.  Patients who developed AKI were noted to have significantly higher levels of D-Dimer on day 1 (hospital admission) and of CRP on day 2 (Table. 3).  These results are consistent with the known interaction of SARS-CoV-2 with ACE2 on vascular endothelial cells in nearly all organs of the body, leading to excessive circulating levels of angiotensin 2, a pro-inflammatory, pro-thrombotic, and vasoconstrictive molecule."

(lines 210-218)

  1. Please indicate the reason for cox proportional hazards analysis and forward variable selection.

We used Cox proportional hazards regression with forward variable selection to determine the effects of risk factors found to be statistically significant on univariate analysis.  Forward variable selection is a type of stepwise regression model in which variables are added one by one.  In each forward step, the one variable that is added is the single variable that gives the single best improvement to the model.  The proportional hazards assumption that risk of death is uniform throughout the duration of the study was confirmed using log–log survival plots.

Minor comments:

  1. Please spell out SOFA and CPR in the abstract.

The abbreviations have been defined.

  1. Please remove the P, OR, and 95% CI values from the abstract.

We prefer to include the statistics in our Abstract.  We will defer to editorial discretion in this matter.

  1. Line 153, please include a comma between the number 3 and IQR.

A comma has been added.

  1. Please add a note to table 3 indicating the meaning of numbers in parentheses.

The following footnote has been added to Table 3.

"Biomarker levels are expressed as median with interquartile ranges (IQR) in parentheses."

  1. Provide a rationale in the text that why coagulopathic markers are an essential risk factor that was necessary to summarize in table 3.

The rationale is provided in our expanded discussion of Table 3, as requested above by the same Reviewer.

Round 2

Reviewer 1 Report

the manuscript has been significantly improved and can be accepted in the present form.

Reviewer 3 Report

The authors have improved the manuscript; I congratulate them on their work.